# The Future of Cancer Diagnosis, Treatment and Surveillance: A Systemic Review on Immunotherapy and Immuno-PET Radiotracers

**DOI:** 10.3390/molecules26082201

**Published:** 2021-04-11

**Authors:** Virginia Liberini, Riccardo Laudicella, Martina Capozza, Martin W. Huellner, Irene A. Burger, Sergio Baldari, Enzo Terreno, Désirée Deandreis

**Affiliations:** 1Department of Medical Science, Division of Nuclear Medicine, University of Torino, 10126 Torino, Italy; desiree.deandreis@unito.it; 2Department of Biomedical and Dental Sciences and of Morpho-Functional Imaging, Nuclear Medicine Unit, University of Messina, 98125 Messina, Italy; riclaudi@hotmail.it (R.L.); sergio.baldari@unime.it (S.B.); 3Department of Nuclear Medicine, University Hospital Zurich, University of Zurich, 8006 Zurich, Switzerland; martin.huellner@usz.ch (M.W.H.); irene.burger@usz.ch (I.A.B.); 4Molecular & Preclinical Imaging Centers, Department of Molecular Biotechnology and Health Sciences, University of Torino, Via Nizza 52, 10126 Torino, Italy; martina.capozza@unito.it (M.C.); enzo.terreno@unito.it (E.T.); 5Department of Nuclear Medicine, Kantonsspital Baden, 5004 Baden, Switzerland

**Keywords:** immune checkpoint inhibitors, immune checkpoint radiolabeled antibodies, PD-1, PD-L1, immune PET, immunotherapy, AI, radiomics, deep learning, CAR-T cells

## Abstract

Immunotherapy is an effective therapeutic option for several cancers. In the last years, the introduction of checkpoint inhibitors (ICIs) has shifted the therapeutic landscape in oncology and improved patient prognosis in a variety of neoplastic diseases. However, to date, the selection of the best patients eligible for these therapies, as well as the response assessment is still challenging. Patients are mainly stratified using an immunohistochemical analysis of the expression of antigens on biopsy specimens, such as PD-L1 and PD-1, on tumor cells, on peritumoral immune cells and/or in the tumor microenvironment (TME). Recently, the use and development of imaging biomarkers able to assess in-vivo cancer-related processes are becoming more important. Today, positron emission tomography (PET) with 2-deoxy-2-[^18^F]fluoro-D-glucose ([^18^F]FDG) is used routinely to evaluate tumor metabolism, and also to predict and monitor response to immunotherapy. Although highly sensitive, FDG-PET in general is rather unspecific. Novel radiopharmaceuticals (immuno-PET radiotracers), able to identify specific immune system targets, are under investigation in pre-clinical and clinical settings to better highlight all the mechanisms involved in immunotherapy. In this review, we will provide an overview of the main new immuno-PET radiotracers in development. We will also review the main players (immune cells, tumor cells and molecular targets) involved in immunotherapy. Furthermore, we report current applications and the evidence of using [^18^F]FDG PET in immunotherapy, including the use of artificial intelligence (AI).

## 1. Introduction

The origins of immunotherapy date back to the discovery of smallpox vaccination by Edward Jenner in 1796. Scientific interest in the mechanisms of immune surveillance of diseases has been high since then. In the last years, several findings on the involvement of both innate and adaptive immunity in the mechanisms of immune surveillance of cancer led to the emergence of immunotherapy, which is today considered one of the most promising anti-cancer approaches [1]. The knowledge of the complex and dynamic interactions between a tumor and the immune system has been fundamental for the development of increasingly specific, personalized and targeted therapies. This was facilitated by the progress of modern medicine through increasing multidisciplinary and translational approaches, including omics sciences (e.g., genomics, proteomics, metabolomics). The same knowledge is also exploited by the growing field of molecular imaging through the identification of diagnostic biomarkers to support clinical decision-making and patient selection by non-invasive, in-vivo assessment of specific therapy target expression, therapy response and toxicity. In recent years, also technological development assumed an important role in supporting clinical decision-making and selecting patients as immunotherapy candidates. First, new digital tomographs have greatly improved the detectability of small lesions, but also the use of artificial intelligence (AI), may improve the interpretation of so-called “big-data” extracted from images and allow for their integration with other medical data. Here, we will review the state of the art and the perspectives in the aforementioned aspects in the field of immunotherapy, with a special focus on molecular imaging. 

## 2. Material and Methods

We searched the PubMed, PMC, Scopus, Google Scholar, Embase, Web of Science, and Cochrane library databases (between January 2015 and February 2021), using the following both as text and as MeSH terms: “immunotherapy”, “ICI*”, “PET”, “micro-PET”, “PET/CT”, “PET/MR”, “convolutional”, “neural”, “network”, “machine”, “learning”. No language restriction was applied to the search, but only articles in English were reviewed. The systematic literature search returned 822 articles. According to the PRISMA flow-chart, after duplicate removal, 86 articles have been considered, fully read, analyzed, and extensively described according to their title and abstract as previously described [2]. We also checked for further relevant articles in the references of the articles included in the retrieved literature. 

## 3. Tumor Microenvironment

As recently observed, cancer should not be considered as a simple collection of transformed cells, but as an organic and evolving complex of transformed tumor cells and other cellular and molecular components that together constitute the *tumor microenvironment* (TME) [1], as illustrated in Figure 1. 

On the one hand, tumor cells and stromal cells, such as cancer-associated fibroblasts (CAFs), adipocytes, and vascular endothelial cells, co-evolve and grow within a protective extracellular matrix (ECM) with the help of cytokines, glycoproteins, proteoglycans and growth factors. On the other side, tumor-infiltrating immune cells present in the TME try to identify and destroy growing tumor cells. This defensive behavior may also be associated with tissue inflammation and a pro-tumorigenic effect responsible for the selection of more aggressive tumor clones, able to block and evade the same immune defensive mechanism [3]. The two major players of the anti-tumorigenic effect of the immune system are the natural killer (NK) cells and the tumor-infiltrating lymphocytes (TILs), which recognize and kill the tumor cells through different mechanisms. First NK cells target cells that do not present the major histocompatibility class I (MHC-I) on their surface. Once activated, NK cells release cytolytic granules to trigger programmed cell death. Differently, CD8^+^ T-lymphocytes and CD4^+^ T helper lymphocytes 1 (Th1) are activated by antigen-presenting cells (APCs), such as dendritic cells (DCs), but also by mononuclear phagocytes and B cells. The APCs can recognize and capture extracellular proteins from tumor cells and present them as antigen-derived peptides to CD4^+^ and CD8^+^ T-lymphocytes by the major histocompatibility complex molecules of class I and II (MHC-I and MHC-II) expressed by these cells; this is a critical step for an effective adaptive immune response [4]. Moreover, CD4^+^ Th1 T-lymphocytes destroy tumor cells also trhough the secretion of cytokines and chemokines, especially interleukin 2 (IL2) and interferon-gamma (IFNγ), which help the recruitment of further NK cells, CD8^+^ T-lymphocytes and macrophages. These mecchanisms also prevent metastatic spread of the tumor [5]. 

Second, tumor-infiltrating immune cells play a more ambivalent role and may as well have an anti-tumorigenic or a pro-tumorigenic effect based on a delicate balance of different mechanisms within the TME. Namely, other TILs can exhibit a pro-tumorigenic effect, such as CD4^+^ T helper lymphocytes 2 (Th2), producing interleukins with an inflammatory effect, and the regulatory CD4^+^ T-lymphocytes (Treg), which are indispensable for maintaining homeostasis, down-regulating the CD8^+^ T-lymphocyte activity [1,6]. Also, the presence of tumor-associated macrophages (TAMs) is related to a poor prognosis and reduced overall survival (OS). TAMs phenotype can be polarized by the TME characteristics, resulting in the development of pro-inflammatory macrophages (M1 type, driven by IFNγ and tumor necrosis factor-alpha (TNFα) or anti-inflammatory macrophages (M2 type, driven by IL4 and IL13). While the former ones have an anti-tumorigenic effect, the latter ones promote tumor progression by stimulating angiogenesis, remodeling the ECM, promoting metastasis and immunosuppression [1,3,5,7,8]. Tumor-associated neutrophils (TANs) may also show a similar behavior: they can be polarized, resulting in the development of an anti-tumorigenic type (N1) or a pro-tumorigenic type (N2). Driven by the action of transforming growth factor-beta (TGF-β) secreted by the CAFs, N2 stimulate cancer proliferation and migration through the secretion of matrix metalloproteinases (MMPs) and IL1β [3,4,9]. Finally, the role of tumor-infiltrating B cells (TIBs) is even more controversial: CD20^+^ TIBs have been shown to behave as APCs with an anti-tumorigenic effect in non-small-cell lung cancer (NSCLC), ovarian cancer [10,11] and in melanoma, as part of the tertiary lymphoid structures (TLS) associated with CD8^+^ T-lymphocytes [12]. Moreover, CD20^+^ TIBs can produce granzyme B and perforin, which directly leads to the activation of apoptotic pathways in tumor cells. In contrast, however, Shalapour et al. demonstrated that TIBs seem to inhibit the anti-tumorigenic effect of CD8^+^ T-lymphocytes, promoting tumor cell growth and migration [13].

## 4. Immunotherapies Targets

The understanding of the complex mechanisms regulating immune cell infiltration in the TME is still evolving. A comprehension of this intricate system is crucial in order to develop new immunotherapy strategies for cancer treatment, such as immune checkpoint inhibitors (ICIs), adoptive cell transfer (ACT), oncolytic virus therapies, cancer vaccines and cytokine therapies (Figure 2).

### 4.1. Immune Checkpoint Inhibitors

ICIs are nowadays the most commonly used immunotherapy strategy. ICIs represent a class of monoclonal antibodies (mAbs), able to counteract the immunosuppressive action of the immune checkpoint (IC) proteins and re-establishing the anti-tumorigenic function of the tumor-infiltrating immune cells present in the TME [1,3,5,14]. As the above-mentioned Treg, ICs are fundamental for maintaining homeostasis, down-regulating CD8^+^ T-lymphocyte activity, and for preventing autoimmune reactions [15]. Cancer cells have learned to take advantage of this mechanism, producing and overexpressing ICs on their cell surface and down-regulating the immune system [16]. Cytotoxic T-lymphocyte associated protein 4 (CTLA-4), programmed cell death protein 1 (PD-1) and its ligands (PD-L1 and PD-L2) are the protagonists of this process. CTLA-4 is a glycoprotein expressed on the T cell surface, which allows tumor cells to escape the immune system by binding to CD80/86 on APCs. Anti-CTLA-4 therapies, such as ipilimumab, block this mechanism [17]. Furthermore, tumor cells can express PD-L1 and PD-L2 on their surface, thereby blocking the activity of T cells, B cells and NK cells, binding their surface receptor PD1. Anti-PD1 therapies, such as nivolumab and pembrolizumab, and anti-PD-L1 therapies, such as avelumab, atezolizumab and durvalumab block this mechanism [18].

### 4.2. Adoptive Cell Transfer

Another widely used immunotherapy technique is the adoptive cell transfer (ACT), whose main representatives are chimeric antigen receptor (CAR)-T cells. CAR-T cells are autologous isolated T cells, genetically engineered through the use of antibody fragments to carry a receptor capable of recognizing specific antigens expressed by tumor cells, being reinfused to patients to stimulate cancer immune surveillance through secretion of perforin and granzyme granules and activation of programmed cell death.

CARs have been categorized into four generations according to the number of intracellular signaling molecules: while the first-generation CAR-T cells induced immune system reaction against target cancer cells, but could not promote CAR-T cells expansion in-vivo following reinfusion (could be reproduced only ex-vivo), on the other hand, other generations of CAR-T cells contained additional intracellular co-stimulatory domains, which allowed these CAR-T cells to grow, expand and ultimately be persistent in the patient’s body [19,20,21].

CAR-T cells were developed for the treatment of hematologic disorders, such as acute lymphoblastic leukemia (ALL) and chronic lymphocytic leukemia (CLL) in adult patients [22,23,24]. Their use for solid cancer is still challenging due to the higher heterogeneity of tumoral antigens, the resulting increased difficulty in configuration recognition, higher difficulty to penetrate tumor tissue through the vascular endothelium and a low survival in the TME [21].

### 4.3. Oncolytic Viruses, Cancer Vaccines and Cytokine Therapies

Another immunotherapy technique uses oncolytic viruses described as modified viruses aimed to target and kill tumor cells. In 2015, the Food and Drug Administration (FDA) approved talimogene laherparepvec (T-Vec), also known as Imlygic^®^, the first oncolytic modified herpes simplex virus, that infects and destroys melanoma cells in patients with unresectable metastatic disease [25,26].

Cancer vaccines are another approach in immunotherapy, relying on tumor-specific antigens to trigger T-cell activation in the TME. Its largest field of application is melanoma (MZ2-E, MZ2-D and gpl00 antigens [27,28,29]), but this technique has also been used in prostate cancer (sipuleucel-T, a dendritic cell-based cancer vaccine [30]).

Despite the efficacy and initial enthusiasm for the use of cytokine therapies, especially in patients with chronic myeloid leukemia and melanoma, IL2 and TNFα have recently been sidelined in favor of the abovementioned more promising ICIs and ACT approaches, owing to their poor tolerability and severe toxicity [1,3,5,14].

### 4.4. Open Question on the Use of Immunotherapies Targets

Among all the aforementioned immunotherapy-based treatment, ICIs appear to be the most rapidly evolving ones. Since the first publication on the use of ipilimumab in a clinical trial in patients with metastatic melanoma in 2010 [31], ICIs have been approved by the FDA for the treatment of several cancer types besides malignant melanoma, such as renal cell carcinoma, squamous cell carcinoma of the head & neck, Merkel cell carcinoma, hepatocellular carcinoma, cervical cancer, small-cell lung cancer, non-small-cell lung cancer, triple-negative breast cancer, gastric carcinoma, urothelial cancer and Hodgkin lymphoma [32].

However, only a subset of patients responds to immunotherapy, with a very heterogeneous response rate depending on ICIs and cancer type [33,34,35,36]. Moreover, immunotherapy is not exempt from autoimmune-like reactions and immune-related adverse events (irAEs), such as skin rash, pneumonitis, colitis, hepatitis and thyroiditis, that occur in approximately 50% of patients overall, and in 14% as grade 3–4 side effect, according to the Common Toxicity Criteria for Adverse Events (CTCAE) [18].

The exact reasons for response heterogeneity, tumor relapse and side effects are still not clear. A high immunohistochemical (IHC) staining of PD-L1 tumor expression [37,38] and/or PD-1 and CTLA-4 TILs expression [39] on biopsy specimens is the first fundamental step for treatment selection, being generally but not only related to higher response rate to ICI therapy. However, evaluation of ICs expression is generally assessed by a single-tumor biopsy, leading to its potential underestimation [40], as tumor heterogeneity is both spatial (inter- and intra-tumoral heterogeneity of ICs expression) and time-related (more aggressive cell clones developing over time) [41,42]. Thus, while multiple-lesion biopsy sampling is not feasible most of the time, grading heterogeneity among primary and metastasis would be a valuable approach trough whole body target expression tecniques.

In clinical practice, the indication for treatment with ICIs is based on the expression of ICs on tumor, assessed on the histological sample of the tumor. Patient selection criteria for therapy with ICIs (percentage of expression of ICs) vary depending on the tumor under investigation and the drugs considered. However, a good overall survival after treatment with ICIs was surprisingly found even in patients with low PD-L1 expression (PD-L1 expression of 1%) on the biopsy sample. For this reason, new criteria for the selection of patients who could benefit from these therapies have been recently evaluated. In particular, the combined positive score (CPS) has been introduced, a composite of ICs (such as PD-L1) tumor proportion score (TPS) and host- and tumor-related parameters, which evaluates the percentage of positive ICs (tumor cells, lymphocytes and macrophages) present in the TME [43,44].

## 5. Molecular Imaging

Non-invasive in vivo imaging techniques, such as computed tomography (CT), magnetic resonance imaging (MRI), positron emission tomography (PET) and single-photon emission computed tomography (SPECT) as single or hybrid modalities (i.e., PET/CT, PET/MRI, SPECT/CT) may provide valuable information for staging, patient selection, treatment response assessment, restaging and follow-up, even though optimal biomarkers to predict immunotherapy response and toxicity have not been identified yet [34].

### 5.1. Immuno-PET

#### 5.1.1. Critical Issues in the Development of a Good Radiotracer in Immunotherapy

The development of a new molecular imaging probe to answer a specific biological question is a complex process, that requires a multidisciplinary team, significant funds and several decisional stages from the laboratory bench to animal models, and from animal models to the clinic. A good molecular probe needs to overcome biological problems, such as rapid clearance, metabolism, degradation and too early excretion, non-specific accumulation in non-target tissues, poor perfusion of the tumor and pharmacological delivery barriers [45]. Also, further potential issues related to radioactive decay and toxicity have to be considered. Ideally, the radiotracer: (1) must be composed of a radioactive isotope with an identical behavior as the stable isotopes of the same element, (2) must not change the chemical and physical properties of the biological system in which it will be introduced, (3) must not deviate the normal physiological state of the system under consideration, (4) must be highly specific and/or selective, with high plasma clearance and low plasma protein binding, (5) must have the same chemical and physical characteristics of the unlabeled compound, (6) must have good in-vivo stability, the labelling between the radionuclide and the compound must be strong enough to avoid the detection of free radionuclide, and (7) must have a blood half-time adequate to the specific biological question [46,47].

New radiotracers have been developed in the field of immunotherapy, using both ex-vivo or in-vivo labelling methods. Ex-vivo cell labeling techniques are currently increasing. Here, immune cells are isolated from a patient and incubated with the radionuclide ex-vivo, despite the high fragility (short viability) of the re-injected cells and only preclinical data available, till now. An in-vivo approach would allow the possibility of PET imaging at any time after cell infusion, especially this could be of particular interest in the case of CAR T cell infusion, allowing to verify their arrival and accumulation in the tumor and metastases rather than a simple assessment of their presence in the peripheral blood by serial sampling, as per current procedural lines [48]. The first trials were carried out using SPECT radionuclides, such as [^99m^Tc] labeled hexamethylpropyleneamine (HMPAO) or [^111^In] labeled 8-hydroxy-quinoline (oxine). In 2003, Meidenbauer et al. [49] labeled Melan-A-specific CD8^+^ cytotoxic T lymphocytes with [^111^In]oxine and demonstrated the localization of these radiolabeled cells to metastatic sites 48 h after injection [49]. More recently, several groups [50,51,52] have studied ex-vivo CAR-T cells labeling techniques using [^89^Zr] for PET imaging. The synthesis and labelling steps are similar to those used for the [^111^In]oxine complex, but the use of [^89^Zr] is more promising, owing to the higher spatial resolution, higher sensitivity, and better signal-to-background ratio of PET compared with SPECT, and owing to its comparably long half-life of 3.27 days compared to 2.80 days for 111In, which is helpful to monitor cell distribution in murine models after administration [53]. However, the efflux of [^89^Zr] is the major drawback of this radionuclide, owing to the high radiotoxicity exhibited by free [^89^Zr]. This problem has been solved by Bansal et al., which covalently labeled [^89^Zr] to cell surface proteins of mouse-derived melanoma cells, mouse dendritic cells and human mesenchymal cells, although only 30–50% of the cells were found to be efficiently bound [50]. In-vivo labeling techniques have been easier to implement and are in more advanced research phases with some clinical applications, already. In particular, radionuclide-labeled immunoglobulins G (IgG) antibodies (mAbs), antibody fragments or small proteins may be used to detect in-vivo the expression of ICs (PD-L1, PD-1 and CTLA-4), and/or other key molecules of immune checkpoint pathways and immune responses through SPECT or PET/CT images [15]. The choice of the right antigen expressed only on target cells with the right radionuclide is fundamental to create a good radiotracer: [^111^In], [^89^Zr] and [^64^Cu] with longer half-lives (67.2, 78.4 and 12.7 h, respectively) are better suited to label intact antibodies (mAbs), while [^18^F] and [^68^Ga] with shorter half-lives (109.8 and 67.7 min, respectively) are more suitable for labeling of smaller ligands, such as nanobodies or small proteins [48,54]. The key radionuclides used for the development of new radiotracers in the immunotherapy field are listed in Table 1.

Full-length antibodies (IgG, mAbs) are large proteins with a molecular weight >150 kDa, resulting in low tumor infiltration and long blood half-life: indeed, they are eliminated in two or three weeks by either excretion (being filtered, reabsorbed and metabolized through proximal tubule of the nephron) or catabolism (through intracellular catabolism by lysosomal degradation to amino acids after uptake by pinocytosis or by a receptor-mediated endocytosis process). This demands the use of long physical half-life radionuclides increasing patient’s radiation exposure and requiring image acquisition up to one week after injection. On the contrary, antibody fragments and small proteins are preferred over IgG/mAbs due to their shorter biological half-life. In fact, unbound molecules are rapidly cleared from the circulation through kidney excretion, resulting in high-contrast images with improved tumor-to-background signal ratio [55]. This phenomenon allows the use of shorter half-life radionuclides, such as [^18^F] and [^68^Ga], which are already widely used in clinical practice, and where images are acquired earlier (after 1–2 h), which is a big advantage in clinical routine. Another significant advantage of using smaller imaging agents is their faster diffusion and better penetration, compared to the heavier full-length IgG/mAbs, into dense solid tumors, leading to better intra-tumoral distribution [56,57].

Another interesting new option is the use of bispecific antibodies, which are able to bind both endogenous T cells (binding especially CD3 on T cells) and tumor-specific antigens, bringing them into close proximity and facilitating the induction of immune cell mediated apoptosis [58]. However, the mechanisms of function of these bispecific antibodies are unclear and in particular it is difficult to understand whether these antibodies have higher affinity for tumor cells or T cells and consequently it is difficult to predict their biodistribution. Molecular imaging could accelerate the development of these drugs by obtaining information on biodistribution and target involvement using radiolabeled bispecific antibodies.

Finally, the chelator is also critical to prevent dissociation of the radionuclide from the compound, as is often the case with [^64^Cu], which tends to accumulate in the liver when it dissociates from the chelator [59].

Among all the new immune-cancer related radiotracers created in recent years, those associated especially with immune checkpoints and adoptive cell transfer techniques, and those targeting CD8+ T lymphocytes, which seem to have the most important anti-tumorigenic role in the TME, are the most promising ones. All the radiotracers tested in clinical trials are listed in Table 2.

#### 5.1.2. PD-1/PD-L1 Pathways

As mentioned above, PD-1/PD-L1 pathway is the most widely used immunotherapeutic approach, being also the most studied pathway in pre-clinical and clinical studies. However, different targets, such as antibodies and small molecules, have been considered to identify a good radiotracer.

##### Labelled Antibodies 

Since the development of the first [^64^Cu]DOTA labeled antibody (IgG) PD-1 radiotracer by Natarajan et al. [63] in 2015, being tested on mice bearing B16-10 melanoma tumors, several immune checkpoint radiotracers have been developed both for SPECT and PET imaging. The development of these radiotracers required some changes in the mice-model used in the preclinical laboratory: immunodeficient mice commonly used to study xenograft tumor models do not provide important information on radiotracer distribution in healthy organs, such as liver, spleen, thymus, lung and brown fat. Therefore, immunocompetent mice have been developed to allow studying both tumor and healthy tissue distribution of these new radiotracers [64]. In 2016, Nedrow et al. studied the in-vivo biodistribution of [^111^In]diethylenetriaminepentaacetic acid (DTPA)-anti-PD-L1 antibody in an immunocompetent murine model of melanoma. Authors reported how splenic uptake of the radiotracer affected the tumor uptake and how the co-injection of labeled antibodies and 100-fold of unlabeled antibodies significantly reduced splenic uptake of the radiotracer at 24 h, shifting the concentration of [^111^In]DTPA-anti-PD-L1 in the blood and potentially increasing tumor uptake [65]. Owing to these results, the assessment of the influence of the cold antigen on radiotracer accumulation in the tumor has been introduced also in many other preclinical studies [66,67,68]. Several antigens have been studied for the development of PD-L1 radiotracers, including full-length IgG/mAbs, antibody fragments and small molecules, while only full-length IgG/mAbs have been used for the development of PD-1 radiotracers.

Interestingly, Kikuchi et al. [69] demonstrated that radiotherapy induced a PD-L1 upregulation in tumor, enhancing the efficacy of either treatment alone. They assessed a [^89^Zr] labeled anti-mouse PD-L1 mAb and dynamic immuno-PET/CT imaging of two murine tumor models (head and neck squamous cell carcinoma and melanoma), alone and during anti-PD-1 mAb immunotherapy. PET/CT imaging demonstrated significantly increased radiotracer uptake in irradiated head and neck tumors compared with non-irradiated flank tumors, whilst anti-PD-1 therapy did not have the same effect.

Similar results have been proven also by Christensen et al. [70]. Authors studied the efficacy of the anti-PD-L1 (clone 6E11) conjugated with dibenzocyclooctyne-Desferrioxamine (DIBO-DFO) chelator and radiolabelled with [^89^Zr] ([^89^Zr]DFO-6E11) on NSCLC xenografts and syngeneic tumors models with different levels of PD-L1 in CT26 tumor-bearing mice subjected to external radiation therapy (XRT) in combination with PD-L1 blockade. [^89^Zr]DFO-6E11 detected the differences in PD-L1 expression among tumor models and quantified the increase in PD-L1 expression in tumors of irradiated mice.

##### Radiolabeled Small Molecules

Several small molecules have been studied to overcome the aforementioned problems related to the use of full-length mAbs. Gaochao et al. [71] designed and developed [^68^Ga]-labeled single-domain antibody radiotracer, [^68^Ga]NOTA-Nb109, for specific and non-invasive imaging of PD-L1 expression in a melanoma-bearing mouse model. Also, Rubins et al. [72] evaluated the PD-L1-targeting affibody molecule ZPD-L1_1 as a PET radiotracer in a mouse tumor model of human PD-L1 expression, radiolabeled with either [^18^F]AlF-NOTA or [^68^Ga]NOTA.

In 2017, Chatterjee et al. [59] produced a PET radiotracer labeling the [^64^Cu]DOTAGA with the WL12 peptide, able to bind PD-L1 with high affinity (half maximal inhibitory concentration (IC50) ≈ 23 nM). They evaluated the ability of [^64^Cu]DOTAGA-WL12 to detect PD-L1 expression in-vivo by PET imaging in NSG (non-obese diabetic (NOD) scid gamma) mice, immunodeficient laboratory mice, harboring Chinese hamster ovary (CHO) tumors with constitutive human PD-L1 expression (hPD-L1) and isogenic negative control tumors. The radiotracer was able to early detect tumor PD-L1 expression, within 60 min after administration, however, with increased uptake in the liver due to the dissociation of Cu2+ from the chelator and its subsequent trans-chelation to plasma proteins, such as albumin and ceruloplasmin. Thereafter, Ravindra et al. [73] studied the WL12 peptide labeling with [^68^Ga]DOTAGA. In immunocompetent mice, the percentage of injected dose per gram (%ID/g) at 60 min resulted much higher in hPD-L1 cells (11.56 ± 3.18), with better tumor-to-background contrast, compared to other cell lines (<4.97). Moreover, the uptake of [^68^Ga]DOTAGA-WL12 in the liver was 50–90% less compared to [^64^Cu]DOTAGA-WL12. Finally, Lesniak et al. [74] evaluated the novel [^18^F]FPy-WL12 radiotracer both in-vitro, in six cancer cell lines with varying PD-L1 expression, and in-vivo, where high liver and kidney uptake indicated the need for an optimized labeling strategy to improve the in vivo pharmacokinetics of the radiotracer.

Still in 2017, Meyer et al. [75] used a small high-affinity engineered protein scaffold (HAC-PD1) to create six different HAC-PD1 radiotracer variants based on the use of different chelate, glycosylation, and radionuclide, tested on mice-models with tumors engineered to either be constitutively positive (CT26 hPD-L1) or negative (DmPD-L1 CT26) for human PD-L1 expression. All radiotracers showed an early (60 min) detection of the PD-L1 positive tumors and [^68^Ga]NOTA-HACA-PD1 and [^68^Ga]DOTA-HACA-PD1, the most suitable candidates for translation into the clinic, exhibited promising target-to-background ratios in ex-vivo biodistribution studies.

In 2018, Truillet et al. [76] developed a [^89^Zr]desferrioxamine B (DFO) radiolabeled recombinant human IgG1 (C4) that specifically binds PD-L1. Authors showed how [^89^Zr]DFO-C4 detected PD-L1 antigen on human NSCLC models and prostate cancer models endogenously expressing a broad range of PD-L1 with a linear positive correlation to the grade of PD-L1 tumor expression. The best tumor-to-background contrast was obtained at 48 h after injection. They also demonstrated how standard chemotherapy changed PD-L1 expression on tumor cells and consequently also changed [^89^Zr]DFO-C4 uptake, underscoring the potential utility of serial imaging to measure clinically relevant expression changes over time.

Further, a study performed by Xu et al. [77] compared a novel anti-PD-L1 antibody radiotracer ([^64^Cu]NOTA-MX001) uptake with [^18^F]FDG uptake in mice bearing MC38 (PD-L1 positive) and 4T1 (PD-L1 negative) xenografts. Interestingly, the uptake of [^18^F]FDG in MC38 and 4T1 xenografts was 5.3 ± 0.4 and 6.4 ± 0.6%ID/g at 1 h, while the uptake of [^64^Cu]NOTA-MX001 was 5.6 ± 0.3 and 1.3 ± 0.4%ID/g at 12 h, allowing to verify the tumor’s expression of PD-L1 in contrast to nonspecific [^18^F]FDG uptake.

In 2019, Wissler et al. [78] developed a site-specific *α*PD-L1 antigen-binding fragment (Fab) conjugate with [^64^Cu]NOTA ([^64^Cu]NOTA-*α*PD-L1) for effective and early visualization and mapping of the biodistribution of PD-L1 in two normal mouse models, which could facilitate the elucidation of the roles of a wide variety of immune checkpoint proteins in immunotherapy at 5, 15, and 45 min post-injection. In the same year, Li et al. [79] evaluated [^89^Zr]Df-KN035, the first 79.6 kDa size anti-PD-L1 domain antibody (KN035), to monitor PD-L1 levels in nude mice bearing LN229 xenografts (brain cancer) with a positive expression for PD-L1. They found that LN229 xenografts were markedly visualized from 24 h after injection of [^89^Zr]Df-KN035, with high uptake enduring up to 120 h with a favorable tumor-to-muscle ratio. The same group also evaluated the impact of chemotherapy, radiotherapy or epidermal growth factor receptor (*EGFR*) tyrosine kinase inhibitor (TKIs) effect on the TME (PD-L1 levels) of NSCLC. PET imaging with [^89^Zr]Df-KN035 was performed before and after *EGFR*-TKI gefitinib treatment to evaluate PD-L1 expression, observing that the high dose of gefitinib inhibited tumor growth and decreased the tumoral uptake of [^89^Zr]Df-KN035 [80].

Finally, another [^18^F]labeled small molecule inhibitor, [^18^F]LN, was designed in 2020 by Yinxing et al. [81] to evaluate PET imaging in both PD-L1 transfected (A375-hPD-L1) and non-transfected (A375) melanoma-bearing mice. Authors observed that tumor uptake (1.96 ± 0.27 %ID/g) reached the maximum at 15 min in the positive group, being 2.2-fold higher than in the negative (0.89 ± 0.31 %ID/g) or the blocked (1.07 ± 0.26 %ID/g) group.

##### Radiolabeled Atezolizumab

Other anti-PD-L1 mAbs have been investigated in several clinical trials, such as [^89^Zr]avelumab in a breast cancer mice-model [66] and the promising [^89^Zr]DFO-atezolizumab in a human tumorgraft model. In 2019, Vento et al. [82] showed for the first time the expression of PD-L1 in a renal cell carcinoma removed from a patient (with favorable nivolumab response) and implanted orthotopically into NOD/SCID mice. This was demonstrated using in-vivo PET images of atezolizumab (monoclonal anti-PD-L1 antibody with a mutant Fc) radiolabeled with [^89^Zr]DFO. Also in 2019, Bensch at al. [61] presented the first-in-human study of [^89^Zr]atezolizumab immune-PET in 22 patients with metastatic bladder cancer, NSCLC, or triple-negative breast cancer. They found a better correlation between progression-free survival (PFS) and OS with pre-treatment radiotracer uptake, compared to conventional IHC staining of PD-L1. These findings highlight the limitations of a single biopsy evaluation and show the benefit of in-vivo assessment by imaging. As in preclinical studies, 10 mg of unlabeled atezolizumab was administered together with the radiotracer to prevent rapid clearance during imaging, which was performed on days 4 and 7. Physiological biodistribution has been reported as following: low uptake in healthy brain, subcutaneous tissue, muscle, compact bone, and lung; higher uptake (increasing over time) in bone marrow, intestines, kidney, and liver. Overall, the observed [^89^Zr]atezolizumab uptake in lymphoid tissue (lymph nodes and spleen mainly) might serve as a surrogate for the activation state of the body’s immune system or as a measure for abundant PD-L1 expression. Tumor [^89^Zr]atezolizumab uptake was generally high, with an overall geometric mean SUVmax of 10.4, but often with a heterogeneous intratumoral, intertumoral and interpatient radiotracer distribution.

##### Radiolabeled Pembrolizumab

In 2020, different targets have been individualized for the development of new immuno-PET radiotracers imaging PD-1 expression. In-vivo pharmacokinetics and whole-body distribution of [^89^Zr]labeled PD-1 targeting pembrolizumab with PET in humanized mice have been recently studied by van der Veen et al. [83], with disappointing results: tumor uptake of [^89^Zr]pembrolizumab was lower than uptake in normal lymphoid tissue, but higher compared to other organs. High uptake in lymphoid tissues (such as spleen, lymph nodes and bone marrow) was reduced by excess unlabeled pembrolizumab administration, which inversely did not affect tumor uptake. Instead, Wenping et al. [84] evaluated the biodistribution of [^89^Zr]labeled pembrolizumab, a humanized IgG4 kappa monoclonal antibody targeting PD-1, in healthy cynomolgus monkeys as a translational model of tracking PD1-positive immune cells with better results: distribution in lymphoid tissue such as mesenteric lymph nodes, spleen, and palatine tonsils increased over time, although it was reduced when a large excess of cold pembrolizumab was co-administered with the radiotracer. Moreover, except for the liver, low radiotracer distribution was observed in all non-lymphoid tissues including lung, muscle, brain, heart, and kidney.

##### Radiolabeled Nivolumab

The early studies with nivolumab were fraught with problems. Similarly to [^64^Cu]DOTAGA-WL12, Cole et al. [85] reported the presence of consistent high liver uptake concerning the in-vivo distribution of [^89^Zr]labeled nivolumab in healthy non-human primates (NHP), but also the possibility to block splenic uptake by co-administration of excess cold nivolumab. England et al. [86] developed another [^89^Zr]nivolumab radiotracer labeled with p-SCN- DFO chelator and a humanized murine model of lung cancer suitable for immuno-PET images. However, in their study, a proper accumulation of [^89^Zr]Df-nivolumab was not reached before 48 h, and uptake values were calculated at 168 h post-injection; moreover, salivary gland and splenic uptake could be another potential complication for proper image interpretation.

In 2018, Niemeijer et al. [60] reported the first-in-human study using whole-body PET imaging with [^89^Zr]nivolumab (anti-PD-1) and [^18^F]BMS-986192, an [^18^F]labeled anti-PD-L1 fibronectin-based protein (adnectin), in 13 patients with advanced NSCLC before treatment with nivolumab. Acquisition time was 7 days after administration of [^89^Zr]nivolumab, reflecting PD-1–expressing tumor-infiltrating immune cells, and 1 h after injection of [^18^F]BMS-986192, reflecting PD-L1 expression in tumor lesions. Both radiotracers showed favorable distribution in tumor, but with a high heterogeneity between patients and lesions, and a high accumulation in the spleen for the presence of immune cells, and in the liver due to catabolism of the radiotracers. Finally, a correlation between radiotracer uptake in tumors and response to nivolumab treatment has found a higher uptake of both radiotracers in responders: [^18^F]BMS-986192 SUV peak (median 6.5 (responder) vs. 3.2 (non-responder), *p* = 0.03), and a similar association was found for [^89^Zr]nivolumab (median SUV peak 6.4 vs. 3.9, *p* = 0.019). Moreover, due to low central nervous system (CNS) tracer penetration, both tracers showed lower accumulation in brain metastases compared to the extracerebral lesions. For non-CNS lesions, the median [^18^F]BMS-986192 SUVpeak was higher for lesions with ≥50% tumor PD-L1 expression by IHC than for lesions with <50% expression. Similarly, [^89^Zr]nivolumab uptake was higher in lesions with a higher aggregates of PD-1 positive tumor-infiltrating immune cells at tumor biopsy.

#### 5.1.3. Cytotoxic T-lymphocyte–Associated Protein 4 (CTLA-4) Pathway

The most important studies on CTLA-4 pathways have been conducted by Ehlerding et al. [87,88], which investigated the capacity to visualize CTLA-4 expression in a NSCLC humanized mouse model by in-vivo [^64^Cu]DOTA-ipilimumab, [^64^Cu]NOTA-ipilimumab and [^64^Cu]NOTA-ipilimumab-F(ab’)_2_ PET images. The visualization of CTLA-4 expression was feasible with all three radiotracers, although all of them demonstrated high absolute uptake in salivary glands, yielding higher salivary gland-to-blood ratios which were slightly lower for F(ab9)_2_ (1.78 ± 0.72 vs. 1.19 ± 0.49, respectively at 48 h). [^64^Cu] labelled ipilimumab (both antibody and fragment) showed a clear salivary gland uptake in humanized mice, and as expected, the fragment showed a faster clearance compared to the entire antibody [87]. Therefore, these radiotracers may help elucidate the response of CTLA-4-targeted checkpoint in immunotherapy, providing a whole-body, in-vivo, non-invasive and quantitative PET imaging of IC biodistribution.

#### 5.1.4. Adoptive Cell Transfer (ACT) Therapy

Different radiotracers for direct and indirect activated cell tracking were developed in the last years [89], such as [^89^Zr]oxine-labelled dendritic cells that showed tumoral uptake at PET imaging of melanoma [90]. Namely, NK cells (designated to eliminate malignant cells [90]) were labelled with an [^89^Zr]oxine complex to perform PET cell tracking in clinically relevant rhesus macaques model for the treatment of hematologic malignancies. Liver, lungs and spleen showed the higher %ID/cc, while a low signal was observed in bone marrow [91]. [^64^Cu]- or [^89^Zr]-labelled antibody against NKp30 were also tested in a renal carcinoma murine model, resulting in an [^89^Zr] higher on-target contrast [92]. Mesenchymal stem cells (MSCs) are multipotent stromal cells that were shown to target tumors, however, their role in neoplastic tissue is still under investigation [93]. To analyze their biodistribution, [^18^F]-labelled Fe_3_O_4_@Al(OH)_3_ nanoparticles were used to track MSC after intravenous injection with combined PET/MRI exam, showing intense liver uptake [94].

#### 5.1.5. Chimeric Antigen Receptor T Cell Therapy (CAR-T)

CD8+ cytotoxic T lymphocytes engineered to express both chimeric antigen receptor (IL-13) and herpes simplex virus type-1 thymidine kinase (HSV1-TK) can entrap the PET radiotracer 9-[4-[^18^F]fluoro-3-(hydroxymethyl)butyl]guanine [^18^F]FHBG within glioma cells. In 2017, Keu et al. [95] showed that non-invasive PET imaging with [^18^F]FHBG can track HSV1-tk reporter gene expression present in CAR-engineered CTLs. In their experience on 7 high-grade gliomas (HGG), [^18^F]FHBG imaging was safe and enabled the longitudinal imaging of T cells stably transfected with a PET reporter gene in patients. Further, the sodium iodide symporter (NIS) has been proposed as a non-immunogenic radionuclide reporter in ErbB T1E28z CAR therapy in two models of triple-negative breast cancer (MDA-MB-231 and MDA-MB-436). The two models are characterized by a different immune checkpoint inhibitor expression. In particular, MDA-MB-231 has a greater expression of PD-L1 compared to MDA-MB-436, and it is inversely correlated with CAR-T tumor retention [96].

A different strategy involves the direct labelling of IL13Rα2-CAR T cells (as described in the ongoing clinical trial NCT02208362) with [^89^Zr]oxine in a glioma mouse model delivered intraventricularly or intravenously [51]. The prostate-specific membrane antigen (PSMA) is a human protein and has a low expression in normal tissue. In a study by of Minn et al., anti-CD19 CAR T cells have been transfected to express PSMA, in order to be tracked with [^18^F]DCFPyL PET in a model of acute lymphoblastic leukemia [97]. UniCAR T cells are modular CAR T cells that can recognize a specific target on the surface of tumor cells. Fused with PSMA-11 epitope, they produce a theragnostic complex that might be useful for retargeting UniCAR T cells in prostate cancer tissue and for treatment response assessment by PET imaging [98].

#### 5.1.6. CD8+ T Lymphocytes

As mentioned above, non-invasive monitoring of active cytotoxic CD8+ T lymphocytes in TME is crucial to predict the immunotherapies’ antitumor responses [99,100]. Their presence has been related to a favorable prognosis in different tumors [43,101].

CD8 is expressed on cytotoxic CD8+ T lymphocytes. It is an antigen highly suitable as a target for immune cell-specific radiotracers [102]. A great effort was made in the last years to optimize radiotracers for detection of CD8+ and CD4+ T cells in TME by immuno-PET imaging. One promising approach is a mini-body labeled with [^89^Zr] that was tested in different immunotherapeutic models, such as colon carcinoma mouse model [102], melanoma, breast, colon, kidney tumor bearing mice [103], gastric adenocarcinoma and Hela cervical cancer [104]. Of note, the minibody [^89^Zr]Df-IAB22M2C has been safely tested in human patients affected by different solid malignancies [62]. As expected, the minibody has a fast blood clearance due to its small size, which allows for early image acquisition. A [^64^Cu]labeled antibody was proposed for the in vivo quantification of CD8+ T cells as a prognostic biomarker in an immunocompetent mouse model of colorectal cancer [105] and breast [106] cancer. Since [^64^Cu] has a shorter half-life (12.7 h) compared to [^89^Zr] (78.4 h), the use of [^64^Cu] reduces patient radiation exposure, thus improving the “acquisition window” compared to shorter-lived PET isotope such as [^18^F] and [^68^Ga]. Rashidian et al. demonstrated that a different CD8+ T cell distribution in TME (homogenous or heterogeneous) can distinguish between responding and non-responding tumors both in melanoma tumor bearing mice treated CTLA-4 therapy [107], and in colorectal adenocarcinoma mouse model treated with anti PD-1 therapy [108]. Namely, in colorectal cancer, both CD8+ and CD11b+ were imaged, and a different distribution was detected in responders and non-responders. A different approach was proposed by Levi et al. using [^18^F]arabinofuranosyl guanine ([^18^F]AraG), a substrate for mitochondrial kinase, which is trapped in activated T-cells [109]. Levi et al. compared its performance as immune response radiotracer compared to [^18^F]FDG in a rhabdomyosarcoma mouse model. Rhabdomyosarcoma, generated by the intramuscular injection of Murine Sarcoma Virus-Moloney Leukemia Virus (MSV-MuLV) is a self-limiting tumor whose rejection, complete by 4 weeks, includes infiltration, activation and proliferation of T cells. As expected, the [^18^F]FDG signal reflected the glucose consumption showing the maximum uptake when the tumor reached the peak of growth, while [^18^F]AraG was found to preferentially accumulate in activated CD8+ cells and with a continuous increase during tumor rejection. Of note, in the colon adenocarcinoma mouse model, responders to anti-PD-1 showed a stronger [^18^F]AraG signal before treatment compared to non-responders [110]. The inducible T-cell costimulatory receptor (ICOS), mainly expressed on activated cytotoxic T cells, memory T cells and regulatory T cells, seem more specific than [^18^F]FDG and [^18^F]AraG [111]. This receptor has been investigated as a PET imaging target for predicting and monitoring T-cell–mediated immune response to cancer immunotherapy. Namely, an [^89^Zr] labeled minibody against ICOS was successfully tested in immunotherapy models of Lewis lung cancer, both in the primary tumor and in tumor-draining lymph nodes. ICOS is an imaging target that allows for an early response detection, before changes in tumor volume occur [112]. In 2017, [^18^F] has been directly introduced to the tryptophane backbone, and radioactive 5-^18^F-L-α-methyl tryptophan was proposed as indoleamine-2,3-dioxygenase 1 (IDO1) imaging agent. IDO1, preliminary tested on melanoma mice, is responsible for the first step of the degradation of the essential amino acid tryptophan into immunosuppressive kynurenine and its inhibition restores the anti-tumour immune response [113]. In 2020, Pandit-Taskar N et al. [62] assessed the safety and utility of [^89^Zr]IAB22M2C, a radiolabeled minibody against tumor-infiltrating CD8-positive (CD81) T lymphocytes, for targeted imaging of CD81 T cells in cancer patients. This first-in-human prospective study included 6 cancer patients (1 melanoma, 4 lung cancer, and 1 hepatocellular carcinoma). Patients received a single dose of [^89^Zr]IAB22M2C (about 111 MBq), followed by PET/CT scans at different timepoints (until 144 h post administration). No side effects were registered, and serum clearance was biexponential. The highest uptake was observed in the spleen, followed by the bone marrow. The maximum uptake in normal lymph nodes was reached between 24–48 h. Uptake in tumor lesions was seen already 2 h after injection, but the uptake in [^89^Zr]IAB22M2C–positive lesions increased until 24 h (SUV from 5.85 to 22.8).

#### 5.1.7. Other Immuno-PET Radiotracers

##### Tumor Associated Macrophages (TAM)

Immunosuppressive macrophages in the tumor microenvironment are associated with poor prognosis. A [^64^Cu]-labelled polyglucose nanoparticle (Macrin) was proposed as a radiotracer for quantitative PET imaging of macrophages in colon adenocarcinoma and in orthotopic lung murine tumor models. The diameter of Macrin nanoparticle is 20 nm, and specific macrophage uptake has been demonstrated by dorsal window chambers. The clinical relevance of this nanosystem is in predicting response, especially to TAM-targeted therapies [114]. Integrin CD11 is expressed on the cell surface of TAM that is associated with poor prognosis in high-grade glioma [115]. An [^89^Zr]-labeled anti-CD11b antibody for non-invasive imaging of TAMs showed significant uptake in glioma compared to contralateral normal brain parenchyma [116]. [^89^Zr]HDL has been proposed as tumor macrophagic nanotracer for PET imaging to evaluate the immunotherapeutic response in a mammary adenocarcinoma model [117]. Goggi et al., tested [^18^F]FEPPA (N-(2-(2-[^18^F]Fluoroethoxy)benzyl)-N-(4-phenoxypyridin-3-yl)acetamide, that binds to TSPO), [^18^F]FDG and two other radiotracers, [^18^F]FB-IL2 and [^68^Ga]mNOTA-GZP (68Ga-NOTA-β-Ala-Gly-Gly-Ile-Glu-Phe-Asp-CHO) that were designed for specific binding of interleukin-2 receptors and granzyme B, respectively. The four radiotracers were directly tested in a murine model of colon cancer, comparing different immunotherapy treatment. While [^18^F]FDG, [^18^F]FEPPA and [^18^F]FB-IL2 were not able to image the immune cell population changes, the [^68^Ga]mNOTA-GZP signal accurately stratified response to combined anti-PD1 and CLTA4 therapy [118].

##### CD3

CD3 is a direct T-cell marker, and the antibody [^89^Zr]DFO-CD3 was shown to predict xenograft tumor model anti-CTLA-4 therapeuty response. Three days after [^89^Zr]DFO-CD3 administration, responders showed higher tumor-to-liver PET uptake compared to non-responders [119]. Also, CD3 has been exploited as a T cell marker in BiTE antibody. BiTE stands for “bispecific T-cell engager”. This category represents special antibodies, which are specific for both a surface target antigen on cancer cells and CD3 on T cells. Their therapeutic efficacy was firstly published in 2009 [120]. Here, an [^89^Zr]-labelled antibody against CEA and CD3 positive cells showed accumulation in colon and breast tumor in PET, and therapeutic property in CEA positive tumors [121]. The full-sized antibody employed in these applications showed long blood half-life and high liver uptake. However, smaller molecules such as minibodies may have more optimal pharmacokinetic properties. A different possible imaging target expressed on activated T cells is the IL-2 receptor, also called CD25 [122]. The PET radiotracer [^18^F]FB-IL2 has been described for the first time in 2012 by Di Gialleonardo et al. [123]. It showed an enhanced lung tumor uptake after irradiation and/or combination with immunization based on the recombinant Semliki Forest (rSFV) viral-vector [124]. An important challenge on the road to [^18^F]FB-IL2 clinical translation is its complex and time-consuming production [125]. To overcome this obstacle, the two variants [^18^F]AlF-RESCA-IL2 and [^68^Ga]NODAGA-IL2, which are characterized by an easier synthesis, were compared to [^18^F]FB-IL2. They showed good in-vitro and in-vivo characteristics, with high uptake in lymphoid tissue and human peripheral blood mononuclear cell (hPBMC) xenografts [65].

##### CD4

CD4+ T cells have also a role in cancer immune response [126]. An anti-CD4 antibody, [^89^Zr]-labeled GK1.5 cDb [127,128] tested in healthy mice reached a strong %ID/g in inguinal lymph nodes, spleen and kidney.

### 5.2. [^18^F]FDG PET/CT

Despite these promising results, the above-mentioned novel radiopharmaceuticals are still all under investigation. In clinical routine, [^18^F]FDG PET/CT has reached a pivotal role in the evaluation of response to immunotherapy, although it is far from a specific immuno-PET radiotracer. [^18^F]FDG allows to visualize rapidly proliferating glucose-avid cancer cells (Warburg effect) and monitor immunotherapy response in different cancer types, such as melanoma, non-small-cell lung cancer, head & neck cancer, urothelial cancer and renal cell cancer [129,130,131,132]. However, during ICIs therapy, the presence of TILs and the inflammatory response may also lead to a transient increase in tumor volume and increased [^18^F]FDG uptake in responding tumoral tissue. This phenomenon, known as pseudo-progression, may potentially lead to an underestimation of the benefit of ICIs therapy: Both morphological Response Evaluation Criteria in Solid Tumors (RECIST) 1.1 and functional PET Response Criteria in Solid Tumors version (PERCIST) 1.0 cannot address pseudo-progression adequately [133,134,135]. Therefore, new morphological and functional criteria have been developed in the last decade to overcome this problem, such as the immune-related response criteria (irRC), the immunotherapy-modified Response Evaluation Criteria in Solid Tumors (imRECIST), the immunotherapy-modified PET Response Criteria in Solid Tumors (imPERCIST), PET Response Evaluation Criteria for Immunotherapy (PERCIMT) and Lymphoma Response to Immunomodulatory Therapy Criteria (LYRIC) [136,137,138,139,140]. The major radiological and functional response criteria to immunotherapy are summarized in Table 3.

In the morphological criteria, both irRC and imRECIST have introduced the category of immune unconfirmed progressive disease (iUPD) for the appearance of new lesions at the first restaging, that must be confirmed at least after 4–8 weeks to be reported as immune confirmed progressive disease (iCPD) [141]. In the functional criteria, the indeterminate response, related to the immune-mediated flare effect, has been introduced also in imPERCIST, PERCIMT and LYRIC criteria. Moreover, the complete disappearance of [^18^F]FDG uptake in all lesions means a complete metabolic response, regardless of a change in tumor size (in contrast to morphological criteria) [142]. Finally, with the intent of differentiating pseudo-progression from true progression, PERCIMT introduced also new criteria of progressive disease (PD), such as the appearance of four or more new lesions (<1.0 cm in functional diameter), or three or more new lesions (>1.0 cm in functional diameter) or two or more new lesions (>1.5 cm in functional diameter) [138]. Compared with morphologic imaging, [^18^F]FDG PET/CT has also the advantage of an early detection of autoimmune reactions and irAEs (such as hypophysitis, pneumonia, colitis, hepatitis, and thyroiditis) due to increased uptake of [^18^F]FDG at these sites. This potentially allows for a rapid intervention in life-threatening cases, and aides in switching therapy in less severe cases [142].

In a recent meta-analysis on the incidence of pseudoprogression during ICIs for solid tumor, Park et al. [143] reported that the incidence according to tumor types were similar for the three more representative ones (6.4% in melanoma, 5.0% in NSCLC, and 7.0% in genitourinary cancer). The incidence of pseudoprogression according to the agent types were also analyzed: the pooled incidence in the studies of PD-1/PD-L1 inhibitor monotherapy was 5.7% (5.6% for PD-1 and 6.8% for PD-L1), whereas one study of melanoma that used a CTLA-4 inhibitor showed higher incidence of pseudoprogression (9.7%).

Several semiquantitative parameters, such as metabolic tumor volume (MTV) and total lesion glycolysis (TLG) rate, have been studied to evaluate the predictive and prognostic value of the metabolic tumor burden at baseline [^18^F]FDG PET/CT. Other parameters, such as spleen to liver ratio (SLR) or bone marrow to liver (BLR) SUVmax ratio, were shown to represent surrogates (indirect index) of the hematopoietic tissue metabolism. Seban et al. [144] in a retrospective study on 55 melanoma patients undergoing 18F-FDG PET/CT before anti-PD1, reported that low tumor burden (TMTV = total metabolic tumor volume; <25 cm^3^) correlates with survival and objective response, while hematopoietic tissue metabolism (BLR > 0.79; SLR > 0.77) inversely correlates with survival.

These results are considered preliminary but may assume a more prominent l role in the evaluation of patients under immunotherapy in the future. Today, the use of [^18^F]FDG PET/CT is not still standardized in clinical practice and cannot not be considered as a specific marker of immune response.

## 6. Artificial Intelligence (AI) in Immunotherapy

An improvement in the acquisition, reconstruction and interpretation of medical image datasets is attempted with the use of AI. Through several approaches (Machine learning, radiomics, and deep learning), the dealing with so-called “big-data” is enabled, with the goals to condense information, to enable better patient selection, to improve workflow and to obtain precise predictive models [145].

### 6.1. Radiomics Assessing the Response to Immunotherapy and Survival Prediction

Radiomic is a new innovative bioinformatic approach to the image’s analysis, that allows to evaluate, through the use of standardized mathematical based models, tumor heterogeneity, quantify predictive and prognostics parameters, radiomic features (RFs), that can be applied in clinical decision support system (CDSS) and in clinical research [146,147,148]. Radiomics features include first-order statistical functions, which do not detect spatial information (such as conventional, histogram, and shape PET parameters), and second-order and high-order statistical functions, which contain information about the spatial relationships between the intensities of more than two voxels (such as texture parameters). In particular, a better assessment and knowledge of inter- and intra-tumor heterogeneity is essential due to his role in resistance mechanism leading to treatment failure.

Valentinuzzi et al. aimed to investigate whether immunotherapy [^18^F]FDG radiomics signature (iRADIOMICS) was able to predict response of metastatic NSCLC (stage IV) to pembrolizumab compared to the clinical standards (iRECIST and PD-L1 immunohistochemistry). 30 patients underwent 18F-FDG PET/CT at baseline, and after one and four months. Multivariate iRADIOMICS was found superior to the current standards in terms of predictive power and survival, with an area under the curve (AUC) of 0.90 vs. an AUCmax of 0.6 for PD-L1 and 0.86 for iRECIST [149]. Polverari et al. evaluating evaluated [^18^F]FDG PET/CT radiomics features extracted from the primary lesion. They were able to predict response to immunotherapy in 57 advanced NCSLC patients They observed that patients with high tumor volume, tumor lesion glycolysis (TLG), and heterogeneity features (“skewness” and “kurtosis”) had a higher probability of immunotherapy failure [150]. Wei et al. assessed 194 patients with advanced NSCLC through a radiomics signature approach (mpRS) to predict response to immunotherapy using an improved least absolute shrinkage and selection operator (LASSO) method, reaching an AUC of 0.81, with similar results for survival prediction [151]. Park et al. tested radiomics features on pre-gefitinib or pre-erlotinib [^18^F]FDG PET/CT images, in order to assess their predictive value in recurrence or advanced NSCLC patients with *EGFR* mutation. Radiomics features were found to have an incremental predictive value of early *EGFR* TKI failure, being also highly predictive for PFS assessment [152]. Parvez et al. aimed to determine if [^18^F]FDG PET/CT radiomics features were able to predict response to immunotherapy and outcome in 82 patients with aggressive B-cell lymphoma. None of the tumor texture features resulted predictive of first-line immunotherapy response, while only a few (i.e., GLNU) correlated with PFS (*p* = 0.013), and OS (kurtosis, *p* = 0.035) [153]. Aide et al. evaluated the prognostic value of [^18^F]FDG PET/CT radiomics features in 132 diffuse large B-cell lymphoma (DLBCL) patients treated with first-line immune-chemotherapy. The feature Long-Zone High-Grey Level Emphasis (LZHGE) reached the highest receiver operating characteristic (ROC) analysis accuracy (0.76), finally resulting as the only independent predictor of 2y PFS [154]. Pridget et al. recently assessed 29 melanoma patients who underwent [^18^F]FDG PET/CT at baseline, and at 1, 3, 6 months after immunotherapy, in comparison with seven biological markers and seven clinical variables. They observed that an increase in spleen-to-liver ratio (SLR) greater than 25% at 3 months [^18^F]FDG PET/CT was associated with poor outcome after immunotherapy [155]. Finally, in a retrospective study on 112 metastatic melanoma patients treated with immune checkpoint inhibition, Basler et al. assessed that a model combined of blood biomarkers (LDH + S100) and non-invasive 18F-FDG PET/CT-based radiomics represent a promising biomarker for early differentiation of pseudo-progression, potentially avoiding added toxicity or delayed treatment switch [156].

### 6.2. Radiomics Assessing the Molecular Profile

Koyasu et al. developed and evaluated a radiomics approach for classifying histological subtypes and *EGFR* mutational status in 156 adenocarcinomas and 32 squamous cell carcinomas based on [^18^F]FDG PET/CT images. Authors used two different machine-learning algorithms (random forest and gradient tree boosting–XGB). For the classification of histological subtypes and *EGFR* mutation status, the highest AUC was obtained with XGB in the multiple types analysis (0.843 and 0.659, respectively) [157]. Zhang et al. assessed the predictive value of pre-immunotherapy [^18^F]FDG PET/CT-based radiomics features for *EGFR* mutational status in 248 NSCLC. Their radiomics model (10 features) was able to discriminate between *EGFR* mutation and *EGFR* wild-type, with an AUCmax of 0.75 for the clinical model, increasing to 0.87 in combination with clinical variables [158]. Also Li et al. aimed to demonstrate the connection between mutations and phenotypes in 115 NSCLC, using somatic mutation testing and [^18^F]FDG PET/CT image analysis. A radiomics signature based on both PET and CT radiomics features outperformed individual radiomics features and the conventional PET parameters in discriminating between mutant and wild-type *EGFR* (AUC of 0.805) [159]. Further, the group of Jiang et al. investigated whether quantitative and qualitative [^18^F]FDG PET/CT features can be used as imaging biomarkers for the *EGFR* mutational status in 80 NSCLC. The 35 selected features were significantly associated with *EGFR* mutational status, and the built predictive model reached an AUC of 0.953 [160]. Jiang et al. explored the potential value of radiomics features from [^18^F]PET/CT in assessing different PD-L1 mutational status in 399 NSCLC. For the prediction of a PD-L1 (Sp142) expression level >1%, a model based on CT features yielded an AUC of 0.97, and AUC 0.97 for [^18^F]FDG PET/CT, as well. For the prediction of PD-L1 (Sp142) expression level >50%, [^18^F]FDG PET/CT features model resulted in an AUC of 0.77 and AUC 0.8 for CT features. [161].

### 6.3. Deep-Learning Approach

Deep-learning (DL) is an AI application organized in multiple, progressive neural networks and subsequent related processes (layers). DL approaches simultaneously learn relevant features and prediction models from input images with no need of previous computation and extraction of “custom-tailored” imaging variables. Park et al. developed a DL model to predict cytolytic activity score (CytAct), using semi-automatically segmented tumors on [^18^F]FDG PET/CT paired with tissue RNA sequencing. In the immune checkpoint blockade (ICB) cohort, a higher predicted CytAct of an individual lesion was associated with more tumor shrinkage after ICB treatment (*p <* 0.001). Furthermore, higher minimum predicted CytAct in each patient was associated with significantly prolonged PFS and OS (*p =* 0.001 and *p =* 0.004, respectively), while in patients with multiple lesions, ICB responders had significantly lower variance of predicted CytActs (*p* = 0.005) [162]. 

## 7. Discussion

As aforementioned, the selection and the response to immunotherapy is based on several mechanisms that play different roles. ICs expression on tumor cells seems to be a necessary, but not sufficient information to to prove sufficient insights into immune mechanisms related to immunotherapy efficay, to select patients candidate to immunotherapy and to predict the reponse to treatment [43,44].

In 2018, Kather et al. [163] introduced the concept of spatial immune infiltration patterns (‘topography’) across cancer entities and across various immune cell types, studying 965 histological tissue slides from 177 patients in a pan-cancer cohort. They showed how a bivariate classification system based on this ‘topography’ can stratify patients and can be considered as a biomarker for patients with solid tumors and candidate to immunotherapy. In a recent review, Pietrobon et al. [164] listed the three distinct “immune topographies”: *hot tumors* characterized by lymphocytes infiltration, mixed with tumor cells in the tumor core; *cold tumors*, characterized by an absence of lymphocytes infiltrations (i.e., almost no lymphocytes can be seen on histological slides); and *immune-excluded tumors* characterized by an abundance of lymphocytes at the invasive edge of the tumor, but few to no lymphocytes in the tumor core. This third type of tumor display gradients of T-cell exclusion, specific to each tumor environment and not present in *cold tumors*, where T cells are completely absent.

ImmunoPET radiotracer, ICO PET radiotracer and [^18^F]FDG give the possibility to systematically mapping these “immune topographies” in both spatial and temporal distribution: - stratifying patient at baseline, before starting a systemic treatment such as immunotherapy or chemotherapy; - providing also important quantitative information, based on conventional and radiomic PET parameters and on dynamic acquisition, that can help to adjust treatment accordingly to immunoPET expression; monitoring “immune topographies” changes during immunotherapies, trough the evaluation of delta-changed of conventional and radiomic PET parameters, helping an early change of therapy in non-responder patients or in case of toxicity.

In-vivo molecular imaging of ICs expression, especially with the use of combined radiotracers, which provide different informations, may be more predictive that in-vitro analsysis of TME. In-vivo molecular imaging provide also the added value of a whole-body evaluation and not limited to a single anatomical sample, leading to the evaluation of the tumor heterogeneity and to the identification of potential non-responding lesions, which need different and/or combined treatment in a personalized medicine approach.

ImmunoPET radiotracers could be moreover useful to detect early signs of drug resistance and to distinguish tumor progression from pseudo-progression, especially if combined with [^18^F]FDG PET/CT images. Moreover, in this contest, ICOS PET radiotracers may have an added value to predict early response to therapy: reasonably, the physician should change therapies regime (switching or combining with other therapies) to the patients, in which ICOS PET shows low or no uptake of activate T cells.

Both immunoPET and ICOS PET seems extremely promising improving clinical patient management, even if ICOS PET seems more versatile than immunoPET and in both cases presumably the information obtained from these radiotracers will still have to be integrated with that extracted from the [^18^F]FDG PET/CT images.

Certainly, functional imaging with PET/CT does not allow analysis of these phenomena at the microscopic level, and biopsy and tissue analysis is still mandatory and strictly necessary. However, immunoPET, ICO and [^18^F]FDG PET/CT images can better guide biopsy for both baseline assessment and restaging in case of disease progression. Combining both the micro- and macroscopic informations, it will be possible to obtain faster a higher level of knowledge about tumor and TME behavior.

## 8. Conclusions

In recent decades, the impressive improvement in translational medicine has led to a better understanding of the TME and all the dynamic mechanisms behind tumor development and growth, such as the relationship between tumor cells and infiltrating immune cells in the TME. These discoveries have led to the development of immunotherapy, and subsequently of imaging biomarkers for the TME, which may provide new insights into TME and cancer behavior. While most of these new radiotracers are still in a preclinical phase, they will certainly provide complementary information to clinically used [^18^F]FDG PET/CT for the prediction of immunotherapy response in the context of personalized medicine, also broadening the theragnostic horizon. Today, [^18^F]FDG PET/CT represent a standard technique employed for tumor staging and response assessment, despite the known limitations due to pseudo progression and enhanced activity of inflammatory components. Therefore, [^18^F]FDG PET/CT is still not validated as a specific biomarker in immunotherapy and not well established for response assessment, yet. Nevertheless, new developments in PET/CT technology and images analysis are becoming fundamental tools to guide therapeutic decision-making and prognosis stratification in clinical practice. AI-based approaches may help identify subpopulations with an increased risk of immunotherapy failure, assessing the immune-profile non-invasively, and predicting survival. Hence, they may well evolve as a fundamental support for clinical decision-making. However, due to problems related to data standardization, privacy concerns, lack of extensive data, the development of AI approaches is still challenging. In our opinion and according to the presented data, new radiotracers combined with technical improvements and AI applications will further refine the selection of the most appropriate therapy for each patient, thus potentially avoiding high-cost strategies (not exempt from toxicity), towards individualized precision healthcare.

## Figures and Tables

**Figure 1 molecules-26-02201-f001:**
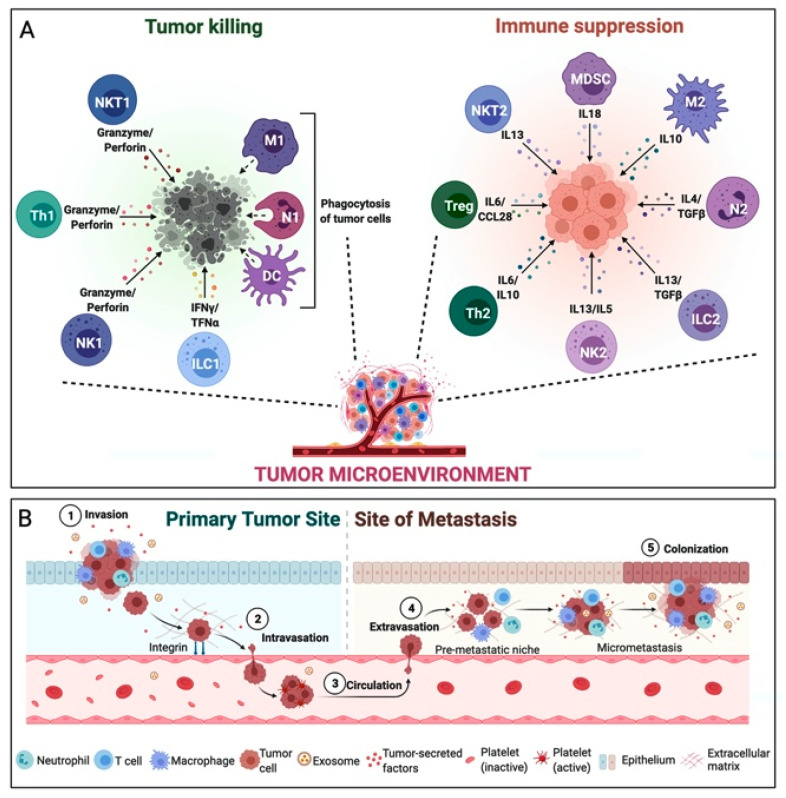
Simplified schematic representation of the tumor microenvironment. (**A**) Schematic representation of immune cells in tumor microenvironment that have an anti-tumorigenic effect (natural killer and tumor-infiltrating lymphocytes) and of other tumor-infiltrating immune cells that have a pro-tumorigenic effect (CD4^+^ T helper lymphocytes 2 (Th2), the regulatory CD4^+^ T-lymphocytes (Treg)). (**B**) schematic representation of immune cells role in tumor spreading process.

**Figure 2 molecules-26-02201-f002:**
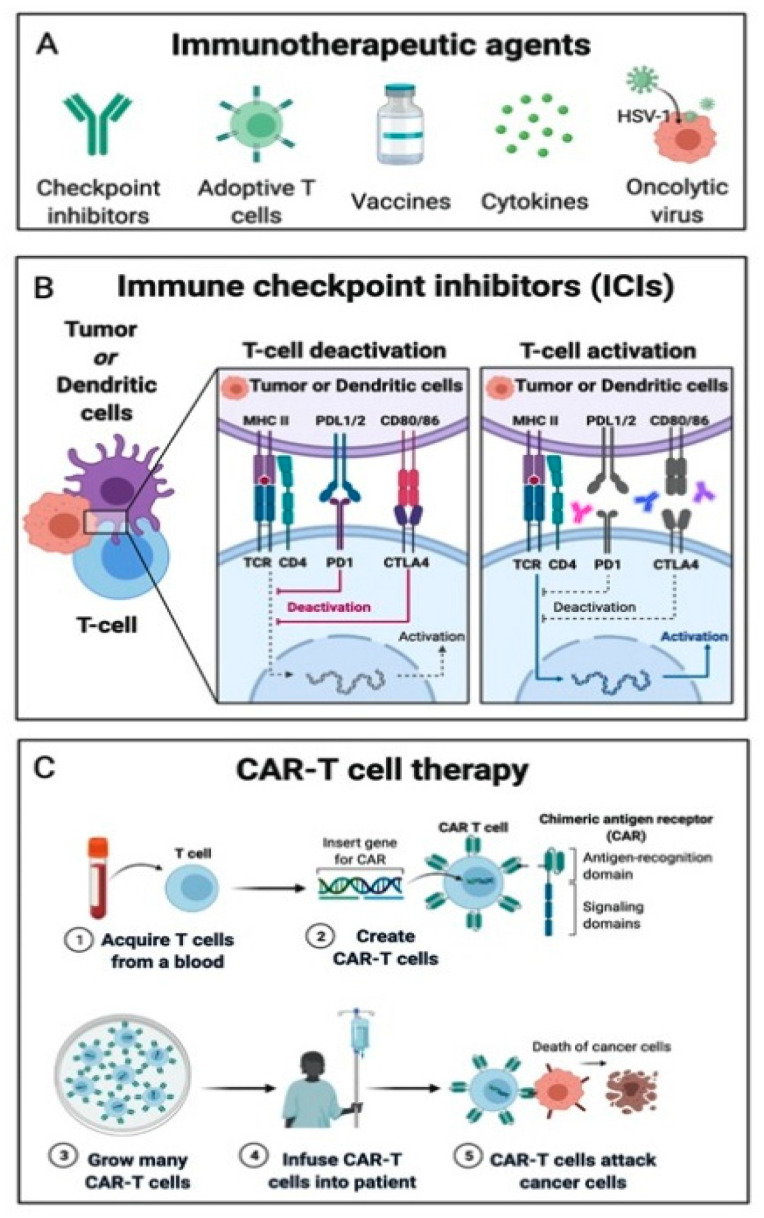
Simplified schematic representation of the new immunotherapy strategies for cancer (**A**), with particular attention to the most widely used immunotherapies: the Immune Checkpoint Inhibitors (**B**) and the CAR-T Cell Therapy (**C**).

**Table 1 molecules-26-02201-t001:** Characteristics of key radionuclide use for the development of new radiotracer in the immunotherapy field.

Radionuclide	Half-life	Type of Emission	Energy of Emission (keV)	Particle Maximum Range in Water
[^11^C] (carbon)	20.4 min	β+	970	3.67 mm
[^68^Ga] (gallium)	67.7 min	β+	1900	9.06 mm
[^18^F] (fluorine)	109.7 min	β+	640	2.16 mm
[^64^Cu] (copper)	12.7 h	β+ (19%) β− (38%) γ (43%)	657 141 511 and 1346	≃ 2 mm
[^89^Zr] (zirconium)	3.27 d	β+	909	≃ 3 mm

**Table 2 molecules-26-02201-t002:** Most relevant radiotracers tested in clinical trials.

Targeting Molecule	Agent	Molecule Type	Tumor Model	Stage	Reference
PD-L1	[^18^F]BMS-986192	adnectin	NSCLC patients	clinical	[60]
	[^89^Zr]-labeled atezolizumab	antibody IgG1	metastatic bladder cancer, NSCLC, or triple-negative breast cancer	clinical	[61]
PD-1	[^89^Zr]nivolumab	antibody IgG4	NSCLC patients	clinical	[60]
CD8^+^	[^89^Zr]Df-IAB22M2C	minibody	melanoma, lung cancer, and hepatocellular carcinoma	clinical	[62]

**Table 3 molecules-26-02201-t003:** Summarized the radiological and functional response criteria to the immunotherapy in tumor.

**Radiological Response Criteria**	**irRC**	**imRECIST**	
Complete response (CR)	Disappearance of all target lesions. Determined by two observations not less than 4 weeks apart.	Disappearance of all target and non-target lesions, without any new lesions. Any pathological lymph nodes must have reduction in short axis to <10 mm. Determined by two observations not less than 4 weeks apart.	
Partial response (PR)	Sum of product of all lesions decreased by >50% for at least 4 weeks; no new lesions; no progression of any lesions.	At least a 30% decrease of the sum of maximum diameters of target lesions; no new lesions; no progression of disease.	
Stable disease (SD)	Sum of product of all lesions decreased by <50% or increased by <25% in the size of one or more lesions.	Does not meet the criteria for CR, PR or PD, taking as reference the smallest sum of maximum diameters of target lesions.	
Progressive disease (PD)	A single lesion increased by >25% (over the smallest measurement achieved for the single lesion) or the appearance of new lesions, that has to be confirmed in 2 consecutive observations at least 4 weeks apart.	Sum of the maximum diameter of lesions increased by >20% over the smallest achieved sum of maximum diameter. The appearance of new lesions and/or progression of non-target lesions are considered iUPD and must be confirmed 4–8 weeks later as iCPD. Progression is not confirmed in case of shrinkage of these lesions at 4–8 weeks and evaluation must be reset.	
**Functional response criteria**	**imPERCIST**	**PERCIMT**	**LYRIC**
Complete metabolic response (CMR)	Complete resolution of [^18^F]FDG uptake within all lesions, to a level of less than or equal to that of the mean liver activity and indistinguishable from the background (blood pool uptake).	Complete resolution of [^18^F]FDG uptake within all lesions, to a level of less than or equal to that of the mean liver activity and indistinguishable from the background (blood pool uptake).	PET Deauville score * = 1, 2, or 3, with or without a residual mass on CT, target nodes/nodal masses must regress to <1.5 cm in longest diameter.
Partial metabolic response (PMR)	Reduction of at least 30% in the sum of SULpeak of all target lesions detected at baseline and an absolute drop of 0.8 SULpeak units.	Reduction of at least 30% in the sum of SULpeak of all target lesions detected at baseline and an absolute drop of 0.8 SULpeak units.	PET Deauville score * = 4 or 5 with reduced uptake compared with baseline and residual mass(es) of any size or on CT > 50% decrease in SPD of up to 6 target measurable nodes and extranodal sites.
Stable metabolic disease (SMD)	Does not meet the criteria for CR, PR or PD.	Does not meet the criteria for CR, PR or PD.	Does not meet the criteria for CR, PR or PD.
Progressive metabolic disease (PMD)	Increased of at least 30% in the sum of SULpeak of all target lesions detected at baseline or new FDG-avid lesions are considered UPMD and must be confirmed 4–8 weeks later as CPMD. Progression is not confirmed in case of PMR or SMD at 4–8 weeks and evaluation must be reset.	Progressive disease if: ≥4 new lesions (<1 cm in functional diameter); ≥3 new lesions (>1 cm in functional diameter); ≥2 new lesions (>1.5 cm in functional diameter).	PET Deauville score * = 4 or 5 with an increase in intensity of uptake from baseline and/or new FDG-avid foci consistent with lymphoma at interim or end of-treatment assessment or on CT > 50% increase in SPD of target measurable nodes and extranodal sites. Immune response exception (IR): - IR1 > 50% increase in SPD in first 12 weeks; - IR2 < 50% increase in SPD with new lesion(s) or >50% increase in PPD of a lesion or set of lesions at any time during treatment; - IR3 = increase in FDG uptake without a concomitant increase in lesion size meeting criteria for PD.

Note: CPMD = confirmed progressive metabolic disease; iCPD = immune confirmed progressive disease; imPERCIST = immunotherapy-modified PET Response Criteria in Solid Tumors; imRECIST = immunotherapy-modified Response Evaluation Criteria in Solid Tumors; irRC = immune-related response criteria; iUPD = immune unconfirmed progressive disease; LYRIC = Lymphoma Response to Immunomodulatory Therapy Criteria; PERCIMT = PET Response Evaluation Criteria for Immunotherapy; UPMD = unconfirmed progressive metabolic disease.* Deauville score: 1 = no uptake above background; 2 = uptake < mediastinum; 3 = uptake > mediastinum but < liver; 4 = uptake greater than liver; 5 = uptake markedly higher than liver (2–3 times in normal liver) and/or new lesions; X = new areas of uptake unlikely to be related to lymphoma.

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
