# Peer review of "The Future of Cancer Diagnosis, Treatment and Surveillance: A Systemic Review on Immunotherapy and Immuno-PET Radiotracers"

_molecules, 2021, doi:10.3390/molecules26082201_

Round 1

Reviewer 1 Report

The manuscript reviews new approaches to imaging that focus on using imaging to support immunotherapy for cancer treatment.  The manuscript is overall well written and makes a solid contribution to an area that has not had much attention. It has done a good job presenting detailed information about what has been published. The prime weakness is that it is not well integrated into bigger questions of the goals of such efforts. The major missing piece is an overview of how the types of imaging under discussion can make a contribution scientifically and therapeutically; how they would be used to make treatment decisions. Why are they better than studies on biopsies? What are the major goals and why are they important?  What are the major challenges?  This missing focus on what is hoped this will accomplish is also missing from the abstract.  This issue should have an early section going into general ideas but also when a molecular target is discussed it should be connected to a reason why getting this in vivo information will have value. 

While a lot of focus is understandable on recognizing the targets for checkpoint blockade, it should also be noted that expression patterns of checkpoint blockade targets is not always, or even generally well correlated with response to the therapies, making a simplistic goal of using it to identify patients that will benefit from the treatments more complex. 

Minor issues

Figure one has 5 distinct figures put together.  The figure is too small and when expanded pixelates so is hard to read.  The legend is very minimal.  While each figure is reasonable self explanatory it still needs support from a more complete legend.  The figure itself is not referred to in the places in the text where it would make sense.  This should be reformatted, possibly into 3 or more separate figures and better explained and integrated into the manuscript

While overall the writing is clear there are poorly phrased sentences that are hard to understand.

poorly phrased lines

Lines 108-10

187-88

Lines 319-20, unclear how using bispecific labeled antibodies brings new value

Lines 527-8, how would the tracers elucidate the response to CTLA4 blockade?

Section on radiomics, line 754 and further.  What is radiomics?

Author Response

Dear Editor,

We would thank you for giving us the chance to revise and improve our manuscript. We also thank the reviewers for their useful comments. We performed a point-by-point revision according to reviewers’ comments. All changes have been highlighted in red font in the text. The manuscript received a linguistic revision by a native speaker.

Virginia Liberini, MD

Reviewer #1:

The manuscript reviews new approaches to imaging that focus on using imaging to support immunotherapy for cancer treatment. The manuscript is overall well written and makes a solid contribution to an area that has not had much attention. It has done a good job presenting detailed information about what has been published. The prime weakness is that it is not well integrated into bigger questions of the goals of such efforts. The major missing piece is an overview of how the types of imaging under discussion can make a contribution scientifically and therapeutically; how they would be used to make treatment decisions. Why are they better than studies on biopsies? What are the major goals and why are they important? What are the major challenges? This missing focus on what is hoped this will accomplish is also missing from the abstract. This issue should have an early section going into general ideas but also when a molecular target is discussed it should be connected to a reason why getting this in vivo information will have value.

 A: We thank the reviewer for his/her comment. According to the reviewer suggestion, we underlined several of the aspect pointed out by the reviewer, even if it must be understanding that the majority of these radiotracer are at on a preclinic stage and it is too early to answer to some of the interesting issues correctly underlined by the reviewer.

1) While a lot of focus is understandable on recognizing the targets for checkpoint blockade, it should also be noted that expression patterns of checkpoint blockade targets is not always, or even generally well correlated with response to the therapies, making a simplistic goal of using it to identify patients that will benefit from the treatments more complex.

A: We thank the reviewer for his/her comment. Based on this and previous comments, we add a discussion session highlighting the radiotracers limit and advantage for each of the three major group of radiotracers (immune checkpoints, adoptive cell transfer and those targeting CD8+ T lymphocytes).

2) Figure one has 5 distinct figures put together. The figure is too small and when expanded pixelates so is hard to read. The legend is very minimal. While each figure is reasonable self-explanatory it still needs support from a more complete legend. The figure itself is not referred to in the places in the text where it would make sense. This should be reformatted, possibly into 3 or more separate figures and better explained and integrated into the manuscript

A: We thank the reviewer for his/her comment. We agree to split the figure into two separate illustration. We have provided two new separate figures in high quality (dpi 300): one for the tumor microenvironment and one for immunotherapy, with a more detailed description.

3) While overall the writing is clear there are poorly phrased sentences that are hard to understand. Poorly phrased lines:

  • Lines 108-10
  • 187-88
  • Lines 319-20, unclear how using bispecific labeled antibodies brings new value
  • Lines 527-8, how would the tracers elucidate the response to CTLA4 blockade?

A: We thank the reviewer for his/her comment. We have paraphrased the sentences indicated by the reviewer to facilitate understanding.

4) Section on radiomics, line 754 and further. What is radiomics?

A: We thank the reviewer for his/her comment. We provide a definition of radiomic from line 812 to 822.

Reviewer 2 Report

This timely review article focuses on immunotherapy treatment for neoplastic cancers. The current challenge in the field is that it is not always clear which patients are most appropriate for this form of treatment, nor are there defined protocols for assessing therapeutic response in individuals undergoing this form of treatment. Current forms of biomarker imaging and assessment are somewhat nonspecific, and therefore this review focuses on emerging technologies for more specific biomarker assessment and analysis, inclusive of artificial intelligence as a tool for predicting therapeutic response. The topic, much like the literature, is broad - however, in order to provide proper treatment of the material for a review, it is necessary to go into some detail, which may not be readily clear to members of the broader scientific community. Having said this, I found this review relevant,  interesting, and comprehensive, but provide the following points for the authors to consider.

  1. The text in figure 1 is difficult for me to discern in my review copy. The authors should provide a higher-resolution image in order to ensure the figure is completely visible.
  2. L187: "reproduced"?
  3. L198: check formatting of space: "In    2005"
  4. L198: "Food and Drug Administration"
  5. L211: "techniques"
  6. L240: correct spacing.
  7. L255: "identical"
  8. L264/265: Please correct this one-sentence paragraph for proper syntax.
  9. L583: "which allows"
  10. L663: please correct "high??"
  11. L797: In most sub-disciplines, genes and mRNAs are expressed in italics, while proteins are expressed without italics. If this sentence references the gene EGFR, please correct here and throughout the manuscript.

Finally, the reviewers should perform an additional round of copy-editing prior to resubmission to ensure proper formatting throughout.

Author Response

Dear Editor,

We would thank you for giving us the chance to revise and improve our manuscript. We also thank the reviewers for their useful comments. We performed a point-by-point revision according to reviewers’ comments. All changes have been highlighted in red font in the text. The manuscript received a linguistic revision by a native speaker.

Virginia Liberini, MD

Reviewer #2:

This timely review article focuses on immunotherapy treatment for neoplastic cancers. The current challenge in the field is that it is not always clear which patients are most appropriate for this form of treatment, nor are there defined protocols for assessing therapeutic response in individuals undergoing this form of treatment. Current forms of biomarker imaging and assessment are somewhat nonspecific, and therefore this review focuses on emerging technologies for more specific biomarker assessment and analysis, inclusive of artificial intelligence as a tool for predicting therapeutic response. The topic, much like the literature, is broad - however, in order to provide proper treatment of the material for a review, it is necessary to go into some detail, which may not be readily clear to members of the broader scientific community. Having said this, I found this review relevant, interesting, and comprehensive, but provide the following points for the authors to consider.

A: We thank the reviewer for his/her comment. Based on the comments of both reviewers, we add a discussion session highlighting the radiotracers limit and advantage for each of the three major group of radiotracers (immune checkpoints, adoptive cell transfer and those targeting CD8+ T lymphocytes).

  • The text in figure 1 is difficult for me to discern in my review copy. The authors should provide a higher-resolution image in order to ensure the figure is completely visible.

A: We thank the reviewer for his/her valuable consideration. We have split the figure into two higher quality figures (dpi 300): one for the tumor microenvironment and one for immunotherapy, with a more detailed description.

  • Minor revision:

L187: "reproduced"?

L198: check formatting of space: "In    2005"

L198: "Food and Drug Administration"

L211: "techniques"

L240: correct spacing.

L255: "identical"

L264/265: Please correct this one-sentence paragraph for proper syntax.

L583: "which allows"

L663: please correct "high??"

L797: In most sub-disciplines, genes and mRNAs are expressed in italics, while proteins are expressed without italics. If this sentence references the gene EGFR, please correct here and throughout the manuscript.

A: We thank the reviewer for pointing out these errors, we have corrected all of them through the text.

  • Finally, the reviewers should perform an additional round of copy-editing prior to resubmission to ensure proper formatting throughout.

A: We thank the reviewer for his/her valuable consideration. We have provided an additional round of copy-editing.

Round 2

Reviewer 1 Report

The authors have endeavored to address prior critiques with mixed results but the overall manuscript is improved.  There are still a variety of poorly worded, grammatically incorrect or unclear sentences, so of which are highlighted below, however the manuscript is generally understandable.

Minor issues

figure 1 and 2 are separated but font in the figure is still very small and impossible to read at 100% size.  There is no value in keeping the font this small.

Line 139: why is the sentence only focused on antigen presentation on class II?  Cross presentation on class I is very important.

Section 283-290 is new writing trying to address utility of expression of targets for immune checkpoint blockade.  The section is not understandable as written

320-21 is a new sentence that is by itself as a paragraph and it is not clear what it is meant to convey as its own paragraph

375-77 is a new sentence that mentions bispecific antibodies but it is not connected to imaging which is the function of the paper. It is described as therapy (inducing apoptosis) which does not fit in this paragraph

915-18 it is not clear how “cold” tumors type 2 differ from immune-excluded tumors

 Discussion of potential value of imaging is now much improved, but the question of resolution needed to determine where in the tumor specific imaged populations exist is not addressed.  The understanding of special resolution of signal from the imaging modalities in vivo needs to be combined with the concept of 3 pattens of immune cell infiltration in tumors and what that may predict. The claims in 919-27 of what can be learned from imaging has to be connected to spatial resolution needed to learn that information, particularly since tumors are not generally simple solid structures but instead integrate normal tissue and tumor cells and tumor stroma.

Author Response

Dear Editor,

We thank you for giving us the opportunity to review and improve our manuscript. We also thank the reviewers for their very nice feedback on the improvements we made based on their valuable suggestions. We performed a point-by-point revision according to the minor comments of reviewer #1 and we hope that these further revisions may meet with your approval. All new changes have been highlighted in red font in the text.

Virginia Liberini, MD

Reviewer #1:

The authors have endeavored to address prior critiques with mixed results, but the overall manuscript is improved. There are still a variety of poorly worded, grammatically incorrect or unclear sentences, so of which are highlighted below, however the manuscript is generally understandable

1) Figure 1 and 2 are separated but font in the figure is still very small and impossible to read at 100% size. There is no value in keeping the font this small.

A: We thank the reviewer for his comment. We have changed the font size of all words in both figures, implementing it by at least 2 sizes. The figures were downloaded from the biorender site in high resolution (dpi 300).

2) Line 139: why is the sentence only focused on antigen presentation on class II? Cross presentation on class I is very important.

A: We thank the reviewer for his/her comment. We agree and we have changed the sentence according to your suggestion.

3) Section 283-290 is new writing trying to address utility of expression of targets for immune checkpoint blockade. The section is not understandable as written

A: We thank the reviewer for his/her comment. We have paraphrased the sentence indicated by the reviewer to facilitate understanding.

4) 320-21 is a new sentence that is by itself as a paragraph and it is not clear what it is meant to convey as its own paragraph

A: We thank the reviewer for his/her comment. We changed the sentence in order to introduce the concept of ex-vivo and in-vivo labelling methods for radiotracer development.

5) 375-77 is a new sentence that mentions bispecific antibodies, but it is not connected to imaging which is the function of the paper. It is described as therapy (inducing apoptosis) which does not fit in this paragraph

A: We thank the reviewer for his/her comment. According to reviewer suggestion, we have better clarified the role of bispecific antibodies and functional imaging in this context.

6) 915-18 it is not clear how “cold” tumors type 2 differ from immune-excluded tumors

A: We thank the reviewer for his/her comment. We provide a better definition of the differences between the two “Immune Topographies”.

7) Discussion of potential value of imaging is now much improved, but the question of resolution needed to determine where in the tumor specific imaged populations exist is not addressed. The understanding of special resolution of signal from the imaging modalities in vivo needs to be combined with the concept of 3 pattens of immune cell infiltration in tumors and what that may predict. The claims in 919-27 of what can be learned from imaging has to be connected to spatial resolution needed to learn that information, particularly since tumors are not generally simple solid structures but instead integrate normal tissue and tumor cells and tumor stroma.

A: We thank the reviewer for his/her comment. According to his/her suggestion, we changed the order of the sentences and added a final paragraph to stress this concept.
